# MP Allosterically Activates AMPK to Enhance ABCA1 Stability by Retarding the Calpain-Mediated Degradation Pathway

**DOI:** 10.3390/ijms242417280

**Published:** 2023-12-08

**Authors:** Hui Li, Mingchao Wang, Kai Qu, Ruiming Xu, Haibo Zhu

**Affiliations:** State Key Laboratory for Bioactive Substances and Functions of Natural Medicines, Beijing Key Laboratory of New Drug Mechanisms and Pharmacological Evaluation Study, Institute of Materia Medica, Chinese Academy of Medical Sciences and Peking Union Medical College, Xian Nong Tan Street 1, Xicheng District, Beijing 100050, China; lhui@imm.ac.cn (H.L.); wangmc@imm.ac.cn (M.W.); qukai@imm.ac.cn (K.Q.); rmxu@imm.ac.cn (R.X.)

**Keywords:** ATP-binding cassette transporter A1, reverse cholesterol transport, AMP-activated protein kinase, MP, calpain-mediated degradation

## Abstract

It is widely recognized that macrophage cholesterol efflux mediated by the ATP-binding cassette transporter A1 (ABCA1) constitutes the initial and rate-limiting step of reverse cholesterol transport (RCT), displaying a negative correlation with the development of atherosclerosis. Although the transcriptional regulation of ABCA1 has been extensively studied in previous research, the impact of post-translational regulation on its expression remains to be elucidated. In this study, we report an AMP-activated protein kinase (AMPK) agonist called ((2R,3S,4R,5R)-3,4-dihydroxy-5-(6-((3-hydroxyphenyl) amino)-9H-purin-9-yl) tetrahydrofuran-2-yl) methyl dihydrogen phosphate (MP), which enhances ABCA1 expression through post-translational regulation rather than transcriptional regulation. By integrating the findings of multiple experiments, it is confirmed that MP directly binds to AMPK with a moderate binding affinity, subsequently triggering its allosteric activation. Further investigations conducted on macrophages unveil a novel mechanism through which MP modulates ABCA1 expression. Specifically, MP downregulates the Cav1.2 channel to obstruct the influx of extracellular Ca^2+^, thereby diminishing intracellular Ca^2+^ levels, suppressing calcium-activated calpain activity, and reducing the interaction strength between calpain and ABCA1. This cascade of events culminates in the deceleration of calpain-mediated degradation of ABCA1. In conclusion, MP emerges as a potentially promising candidate compound for developing agents aimed at enhancing ABCA1 stability and boosting cellular cholesterol efflux and RCT.

## 1. Introduction

Notoriously, atherosclerotic cardiovascular diseases (ASCVDs), including coronary disease, ischemic heart disease, and hypertension, remain the leading cause of morbidity and mortality worldwide [1]. Reverse cholesterol transport (RCT) is the process by which cholesterol is transported from peripheral tissues (including arterial wall foam cells in atherosclerotic plaques) to the liver by high-density lipoprotein (HDL) and then to the bile for intestinal excretion [2]. Previous animal and human studies have confirmed the inverse relationship between RCT and atherosclerosis, with the development of atherosclerosis being delayed when the RCT process is facilitated and, conversely, the impaired RCT process will result in accelerated atherosclerosis [3]. Therefore, it is reasonable to conclude that targeted improvements in the RCT process will slow the progression of atherosclerosis. To achieve this purpose, three conceptual strategies have been proposed: (1) promote macrophage cholesterol efflux, (2) improve the ability of HDL to accept or transport cholesterol, and (3) improve hepatic cholesterol uptake and biliary/intestinal excretion [4]. Of these, macrophage cholesterol efflux is considered the first and rate-limiting step of RCT [5]. Therefore, the most effective strategies for reducing ASCVDs may target macrophage cholesterol efflux in RCT [2,6].

ATP-binding cassette transporter A1 (ABCA1) and ATP-binding cassette transporter G1 (ABCG1) belong to the ATP-binding cassette (ABC) transporter superfamily, were found to be responsible for the mediation of the cholesterol efflux in macrophages [7]. In particular, ABCA1 plays a crucial role in macrophages and is thought to be a primary regulator of cholesterol efflux [8,9]. It promotes the removal of excess cholesterol, thus preventing them from being converted into foam cells and preventing atherosclerosis [7,10]. ABCA1 is one of the target genes of the liver X receptor (LXR), a key sterol-sensitive transcription factor in macrophages that regulates intracellular cholesterol [11]. Surplus intracellular cholesterol induces LXR to bind to the four-nucleotide (DR-4) element of the ABCA1 promoter to drive the ABCA1-mediated efflux pathway, resulting in protection against cholesterol toxicity and maintenance of cholesterol homeostasis [12,13]. In addition to LXR-mediated transcriptional regulation, post-translational regulatory mechanisms also play a vital role in regulating ABCA1 expression [14]. For instance, calpain-mediated proteolytic degradation is considered one of the most important regulatory mechanisms of ABCA1 functions [15]. Calpains, a family of calcium-activated cysteine proteases, are reportedly present in all human tissues and regulated by intracellular calcium concentration [16]. Since calcium ions are essential for various biological processes in all living cells, the levels of free calcium ions in the cytoplasm and extracellular fluid of resting cells must be tightly controlled [17]. There are two major sources of calcium ions in the cytoplasm: the extracellular space and intracellular stores (sarcoplasmic reticulum or endoplasmic reticulum). Voltage-gated calcium channels primarily regulate the former, and the latter is controlled mainly by the ryanodine receptor (RyR) and inositol trisphosphate receptor channels [18,19]. 

On the other hand, a sequence rich in proline, glutamic acid, serine, and threonine (PEST sequence) located in the nucleotide-binding domain 1 (NBD1) of ABCA1 helps to target calpain to ABCA1 on the cell surface, resulting in degradation of ABCA1 by calpain [8,20]. Interestingly, the above effects of the PEST sequence require Thr-1286 and Thr-1305 phosphorylation. Molecules that dephosphorylate the PEST motif can consequently stabilize the ABCA1 located on the plasma membrane [21]. According to the above studies, it is reasonable to conceive that both inhibition of calcium ions in the cytoplasm and dephosphorylation of PEST sequences could retard calpain-mediated proteolytic degradation of ABCA1, increase cellular cholesterol efflux, improve RCT function, and prevent atherosclerosis.

IMM-H007 (2′,3′,5′-tri-O-acetyl-N6-(3-hydroxyphenyl) adenosine), an adenosine derivative derived from cordycepin, has previously been reported to be an agonist of AMP-activated protein kinase (AMPK) that can repress inflammatory atherosclerosis [22]. AMPK is a highly conserved serine/threonine protein kinase widely expressed in higher eukaryotes. Our previous study found that IMM-H007 promotes ABCA1-mediated RCT by inhibiting the calcium-activated calpain activity rather than altering LXR expression, ultimately reducing atherosclerotic plaque formation in hypercholesterolemic mice [23]. Noteworthily, our later investigation manifested that the atheroprotective properties of IMM-H007 depend on the pharmacological activation of AMPK [24]. Moreover, our pharmacokinetic results indicated that IMM-H007 is a prodrug that can be rapidly metabolized to its deacetylated metabolite in the blood by esterase after oral administration, then transported into the cell via equilibrative nucleoside transporter (ENT) and finally phosphorylated to ((2R,3S,4R,5R)-3,4-dihydroxy-5-(6-((3-hydroxyphenyl) amino)-9H-purin-9-yl) tetrahydrofuran-2-yl) methyl dihydrogen phosphate (phosphorylated metabolite, MP) by adenosine kinase [25]. In this case, whether MP is the actual molecule responsible for the promotion of ABCA1-mediated RCT by IMM-H007 and the accompanying mechanisms needs further study.

This study aims to determine whether MP is the actual molecule responsible for the pharmacological activities exerted by IMM-H007 and to elucidate the underlying mechanisms by which MP facilitates ABCA1-mediated RCT function. By virtue of multiple experiments, we have confirmed that MP is the actual active form of IMM-H007. MP stimulates AMPK Thr172 phosphorylation in a concentration- and time-dependent manner, consistent with previous results with IMM-H007. These activations of AMPK are due to the fact that MP can bind directly to AMPK with a moderate affinity and resultantly alter its spatial conformation. Furthermore, subsequent experiments revealed that MP promotes functional ABCA1 expression by inhibiting calpain-mediated proteolytic degradation rather than transcriptional regulation and that this process is dependent on AMPK activation. In more detail, we found that the mechanism by which MP retards calpain-mediated proteolytic degradation is by suppressing calcium-activated calpain activity but not by dephosphorylating the PEST sequence in ABCA1. Collectively, MP is a direct agonist of AMPK and a promising candidate molecule for strengthening ABCA1 stability, increasing macrophage cholesterol efflux, and preventing atherosclerosis.

## 2. Results

### 2.1. MP Facilitates AMPK Thr172 Phosphorylation in a Concentration- and Time-Dependent Manner

The structure of MP is shown in Figure 1A. The biotransformation of the prodrug IMM-H007 to MP was unearthed in our previous work. As shown in Figure 1B, IMM-H007 can be rapidly metabolized into M1 in the blood by esterase after oral administration, and M1 then was transported into the cell through ENT and phosphorylated to MP by adenosine kinase. Cell viability at increasing concentrations (1, 2.5, 5, 10, 25, 50, 100, 200, 400 μM) was detected using the cell-counting kit-8 (CCK-8) to determine the safe dose of MP in current experiments. As shown in Figure 1C, MP showed no cytotoxicity in the 1–400 μM concentration range and even offered a significant increase in cell survival at concentrations above 100 μM, indicating that MP has a wide safety window. Immunoblotting bands revealed that the expression levels of p-AMPK were raised in a concentration-dependent manner (Figure 1D–G) after treatment of J774A.1 and THP-1 cells with MP at different concentrations (0, 0.1, 1, 10, 100, 200 μM). The results were significant when the MP concentrations were higher than 100 μM. In addition, the expression levels of p-AMPK were markedly increased in a time-dependent manner (Figure 1H–K) after treatment of J774A.1 and THP-1 cells with MP for different periods (0, 2, 4, 8, 12, 24 h). These results were consistent with the activation of AMPK by IMM-H007 found in our previous studies [26,27], which suspected that MP was the actual molecule that activates AMPK but not IMM-H007.

### 2.2. MP Is the Actual Molecule That Activates AMPK

To confirm the above hypothesis, we next employed microscale thermophoresis (MST), a technique for rapid quantitative analysis of biomolecular interactions [28], to experimentally uncover the direct interaction between MP and AMPK. As expected, MST findings showed that MP could directly bind to AMPK. By analyzing the typical thermophoretic curve (Figure 2A), the equilibrium dissociation constant (K_d_) value between MP and AMPK was calculated to be 4.27 μM, with a signal noise ratio of 9.8 (S/N > 5 is credible), which was a moderate binding affinity (Figure 2B). Additionally, we applied isothermal titration calorimetry (ITC) to deeply explore the thermodynamic information during the binding of MP to AMPK to elucidate the mechanism of this interaction further and gain insight into the structure-function relationship. According to the titration curve (Figure 2C), we selected the “one set of sites” model to integrate the ITC raw data and fit the binding curve (Figure 2D). Based on this binding curve, the K_d_ value was calculated to be 9.75 μM, which was close to the K_d_ value calculated by MST. This result reinforces the moderate binding affinity between MP and AMPK (10^−6^ M < K_d_ < 10^−5^ M). Further, we calculated the thermodynamic alterations (enthalpy (ΔH), entropy (ΔS), and Gibbs free energy (∆G)) accompanying this binding. As presented in Figure 2E, ∆G, ∆H, and −TΔS were calculated to be −6.84 kcal/mol, −4.19 kcal/mol, and −2.65 kcal/mol, respectively. All these values are less than zero, which means that the binding of MP to AMPK is a process of releasing energy, and the binding of MP to the drug pocket of AMPK is favorable to the overall energy reduction. It maintains the stability of the ligand-protein complex.

### 2.3. Characterization of MP Binding to AMPK

Furthermore, since saturation transfer difference nuclear magnetic resonance (STD-NMR) can screen the binding epitopes of ligands to proteins and determine the ligand protons that interact with the binding site of a protein or the active center of an enzyme [29], STD-NMR has been used to unveil the binding profile of MP to AMPK. As shown in Figure 2F,G, adenosine monophosphate (AMP) and MP bind directly to AMPK. Among them, AMP interacts with AMPK through 2-H and 8-H on the purine ring and 1-H′ and 5-H′ on the pentose ring. MP interacts with AMPK through 2-H and 8-H on the purine ring, 1-H′ and 5-H′ on the pentose ring, and 2-H′′, 4-H′′, 5-H′′, and 8-H′′ on the benzene ring. The above results found that MP possessed more interaction protons (benzene ring) on AMPK than that of AMP, inferring that the binding affinity of MP to AMPK might be more potent than that of AMP under the same condition.

Next, we performed titration experiments to deeply study whether MP and AMP shared the same protein binding site and whether they had a competitive binding relationship. Results showed that with the increase in AMP concentration, the purine and benzene ring signal intensities of MP showed a gradient decrease. In contrast, the signal intensity of the purine ring of AMP increased gradually (Figure 2H). When the molar ratio of MP to AMP was higher than 1:8, the benzene ring signal of MP almost disappeared. The above results confirmed a competitive relationship between MP and AMP during the binding process with AMPK, suggesting they had common binding sites on AMPK.

### 2.4. MP Binding Alters the Secondary Structure Composition of AMPK

Since circular dichroism (CD) has long been considered a valuable tool to investigate whether biomacromolecules such as proteins undergo structural changes due to complex formation [30], we finally employed it to uncover the alterations in secondary structure composition during the process of MP binding to AMPK, including α-helix, β-sheet, β-turn, and random coil. Far-UV CD spectra (190–245 nm) provide information about the secondary structure of proteins. As shown in Figure 2I, AMPK represented two negative peaks around 208 and 222 nm, corresponding to the α-helix of AMPK. The intensity of the negative peaks changed by adding MP, revealing the secondary structure alterations of AMPK. The detailed portions of the secondary structure are exhibited in Figure 2J. When 6 μM MP was added to the AMPK solution, the content of α-helix and β-turn increased (α-helix increased from 11.2% to 19.3%, and β-turn increased from 10.3% to 17.3%). Conversely, the content of β-sheet decreased (from 34.7% for the pure enzyme to 20.9% for 6 μM MP), but the changes in random coil contents are negligible. The results suggested that MP binding would alter the secondary structure of AMPK, resulting in changes in the spatial conformation of the enzyme.

### 2.5. MP Promotes the Expression of Functional ABCA1 by Activating AMPK

After ascertaining that MP was the direct agonist of AMPK and elucidating their binding profile, we further investigated the effects of MP on ABCA1-mediated cholesterol efflux. Immunoblotting results showed that the expression level of ABCA1 in J774A.1 cells sharply increased after treatment with MP at concentrations of 10, 100, and 200 μM for 24 h, or 100 μM for 12 and 24 h (Figure 3A–D). Similarly, MP facilitates the expression level of ABCA1 in THP-1 macrophages (Figure 3E–H). Obviously, the optimal concentration and time to obtain the best activation were 100 μM and 24 h, respectively. Compound C, a specific inhibitor of AMPK, was next used to study whether inactivated AMPK would abolish the promotion of MP on ABCA1. As expected, AMPK suppression dramatically reduced the activation of MP on ABCA1 (Figure 3I–L). Cell surface ABCA1 called functional ABCA1, was demonstrated to be responsible for cholesterol efflux and HDL biogenesis [13]. Therefore, we specifically extracted cell membrane proteins for probing the role of MP on functional ABCA1. As displayed in Figure 3M,N, the expression level of functional ABCA1 in J774A.1 cells was also significantly enhanced by MP treatment, whereas the effects were abolished by compound C intervention. Similar results to J774A.1 cells were obtained with THP-1 macrophages (Figure 3O,P). Compound C repressed the promotion of MP on functional ABCA1. In summary, the above data confirmed that MP regulated intracellular cholesterol efflux by promoting total and functional ABCA1 expression levels and that this regulatory process depended on AMPK activation.

### 2.6. MP Promotes ABCA1 Expression Not through Transcriptional Regulation

Since the upregulation of ABCA1 expression was ascribed mainly to LXRα, a critical upstream promotor for ABCA1 expression [31], we then determined alterations in LXRα protein and mRNA levels. However, compared to the control group, all other groups had no significant difference (Figure 4A,B). The same results were obtained in THP-1 macrophages, where the expression level of LXRα changed negligibly during MP treatment (Figure 4C,D). Simultaneously, changes in LXRα mRNA and ABCA1 mRNA during MP treatment were monitored using RT-PCR experiments. Experimental data found that neither LXRα mRNA nor ABCA1 mRNA changed significantly during MP treatment in both J774A cells and THP-1 macrophages (Figure 4E–H). Based on the current results, it is reasonable to infer that transcriptional regulation is not the pathway by which MP promotes ABCA1 protein expression. The underlying pathway needs further investigation.

### 2.7. MP Delays ABCA1 Degradation through the Calpain-Mediated Pathway

In addition to the transcriptional regulation of ABCA1, its post-translational regulation is equally important [14]. The above paradoxical results prompted us to investigate whether MP affects the ABCA1 post-translational regulation. In this experiment, the protein synthesis inhibitor cycloheximide (CHX) was used to monitor the degradation process of ABCA1 at different time points. As shown in Figure 4I,J, the relative content of ABCA1 in the vehicle group gradually decreased and almost disappeared after 120 min of CHX treatment. However, the degradation of ABCA1 was retarded by MP treatment, and the relative content of ABCA1 did not change significantly throughout the experimental period. These results confirmed the hypothesis that MP activates ABCA1 by delaying ABCA1 degradation but not generation.

A previous study has shown that calpain targets the PEST sequence of ABCA1 to accelerate its degradation [20]. Here, we used increasing concentrations of Ca^2+^ to activate calpain activity to probe whether MP delayed ABCA1 degradation through a calpain-mediated pathway (Figure 4K). As shown in Figure 4L,M, the change in ABCA1 expression levels was negligible after MP treatment, whereas a significant decrease in ABCA1 expression levels occurred in non-MP-treated cells at CaCl_2_ concentrations higher than 10 μM. Additionally, the expression level of calpain was only slightly increased in the presence of MP. However, when the CaCl_2_ concentration was higher than 100 μM and in the absence of MP, the expression of calpain was significantly increased. The results mentioned above suggested that MP can slow down the degradation process of ABCA1 by suppressing calpain activity, ultimately increasing the expression level of ABCA1.

### 2.8. MP Inhibits Calpain Activity by Blocking Extracellular Ca^2+^ Influx

Although the preceding results have confirmed the inhibitory effect of MP on calpain, the underlying mechanism of the inhibition is still a mystery. Thus, we next investigated whether MP reduces intracellular Ca^2+^ levels to inhibit calpain activity. As presented in Figure 5A, the intracellular Ca^2+^ level was markedly increased in the ionomycin group compared to the vehicle group, whereas the increase was reversed after treatment with MP for 24 h. Moreover, we found that 10 μM MP also significantly reduced intracellular Ca^2+^ content (Figure 5B). To better understand the mechanism by which MP reduced intracellular Ca^2+^ content, we then designed experiments to examine the extracellular Ca^2+^ influx and organelle Ca^2+^ release pathways, both of which can cause elevated intracellular Ca^2+^ content. Results showed that the lowering effect of MP on the intracellular Ca^2+^ level was slight after blocking Ca^2+^ influx with SKF96365, inferring that MP reduces intracellular Ca^2+^ levels primarily by inhibiting extracellular Ca^2+^ influx and secondarily through organelle Ca^2+^ release (Figure 5C,D). Meanwhile, our data found that the lowering effect of MP on intracellular Ca^2+^ level persisted after blocking the organelle Ca^2+^ release with 2-APB (Figure 5E–G). These data indicated that MP reduced intracellular Ca^2+^ levels primarily by inhibiting extracellular Ca^2+^ influx. However, the calcium channels on which MP acts remain to be elucidated.

### 2.9. MP Blocks Extracellular Ca^2+^ Influx by Suppressing Cav1.2 Channel

Next, electrophysiological experiments were performed to identify the calcium channels through which MP blocks extracellular Ca^2+^ influx. We first checked the L-type calcium channel Cav1.2. Results showed that the inhibition ratio of 10 μM MP and 100 μM MP on the Cav1.2 channel was calculated to be 7.04% and 33.66%, respectively (Figure 5H,I). In addition to the Cav1.2 channel, the T-type calcium channel Cav3.2 was checked in the following experiments. As shown in Figure 5J,K, the inhibition ratio of 10 μM MP and 100 μM MP on the Cav3.2 channel was calculated to be −1.51% and 12.54%, respectively. These findings supported the view that MP blocks extracellular Ca^2+^ influx mainly by suppressing the Cav1.2 channel.

### 2.10. MP Inhibits Calpain-ABCA1 Interaction Independently of Dephosphorylation of the PEST Sequence

Next, the PEST sequence of ABCA1 was investigated to gain insight into whether MP could attenuate calpain targeting to the PEST sequence by inhibiting the phosphorylation of the PEST sequence, thereby slowing down degradation. As displayed in Figure 6A, the co-immunoprecipitation results confirmed the interaction between calpain and ABCA1, consistent with the previous study [21]. Moreover, by calculating the ABCA1/calpain relative content (Figure 6B), it was found that the ABCA1 content increased in the MP group while the calpain content decreased, inferring that the MP may have attenuated the interaction between calpain and ABCA1, resulting in the opposite alterations. We then investigated whether MP attenuates calpain-ABCA1 interactions by dephosphorylating PEST sequences by identifying phosphorylation modification sites on ABCA1. The total ion chromatograms (TIC) collected by LC-MS/MS were presented in Appendix A. Identification results are listed in Table 1. Unexpectedly, MP treatment promoted the phosphorylation of ABCA1 at sites 1114, 1221, and 1682 but did not affect the phosphorylation status of the PEST sequence (1283–1306), suggesting that MP attenuated the calpain-ABCA1 interaction not by promoting the dephosphorylation of the PEST sequence, but through other potential mechanisms.

## 3. Discussion

To date, atherosclerosis remains a prominent risk factor for various cardiovascular diseases, imposing a substantial burden on human society [32]. This underscores the significance of identifying a candidate molecule with a potential atheroprotective effect. In this study, we present a promising molecule, MP, identified as a phosphorylated metabolite of IMM-H007. In the first half of this study, we observed that MP exhibits a direct and modest binding affinity to AMPK, subsequently promoting its phosphorylation. This finding, consistent with the effects observed with IMM-H007, supports the hypothesis that MP is the active molecule responsible for IMM-H007’s anti-atherosclerotic effects. Furthermore, the latter part of our investigation focused on elucidating the potential mechanism by which MP enhances the stability of the ABCA1 protein following AMPK activation. Specifically, we observed that activated AMPK not only downregulates the Cav1.2 channel, impeding the influx of extracellular Ca^2+^ and thereby reducing intracellular Ca^2+^ levels and inhibiting calcium-activated calpain activity, but also weakens the interaction between calpain and ABCA1. These events culminate in the deceleration of calpain-mediated degradation of ABCA1. More importantly, gaining a deeper understanding of the mechanism by which MP promotes the functionality of ABCA1 holds the potential to provide insights into strategies for the prevention and amelioration of atherosclerosis.

Our prior investigations have established the rapid metabolism of IMM-H007 in the bloodstream, indicating that the biological effects attributed to IMM-H007 are not a result of its own actions. Furthermore, previous studies have independently shown that IMM-H007 effectively activates AMPK and inhibits ABCA1 degradation. However, the causal relationship between IMM-H007's inhibition of ABCA1 degradation and its activation of AMPK remains unclear [23,24]. Building upon these findings, this study was specifically designed to address two fundamental questions: What is the source of the observed effects? And how are these effects generated?

To address the initial question of “What is the source of the observed effects?” we employed pharmacokinetics [25] and molecular biology. In this paper, we unveil the moderate direct interaction between MP and AMPK and the associated biological effects of MP on AMPK activation at both the molecular and cellular levels. We observed that the affinity of MP for AMPK ranges between 1–10 μM, which is significantly lower compared to other direct agonists of AMPK, such as PF-06409577 (0.005 μM) [33], 991 (0.06 μM) [34], A769662 (0.5 μM) [34], but similar to that of AMP (20 μM) [35]. Surprisingly, these direct AMPK agonists with high affinity remain in the preclinical stage, serving as experimental tools. Therefore, we speculate that stronger in vivo activation of AMPK may not necessarily be advantageous. Due to the fact that AMPK has up to 12 protein isoforms and is widely distributed in the body, stronger affinity may potentially lead to more side effects, thereby restricting their further development. In contrast, MP exhibits a comparable affinity to that of AMP, an endogenous agonist of AMPK. Hence, we believe that MP's more moderate binding mode may facilitate its progression into further clinical studies. Collectively, these findings confirm that MP is the source of the observed effects. However, this section of the study has certain limitations. For instance, the biophysical methods utilized to characterize the interactions between MP and AMPK were rudimentary. The exhaustive conformational alterations accompanying this binding process and the structure-function relationships are areas that require further elucidation through the application of advanced structural biology techniques.

Secondly, we addressed the second question “how are these effects generated?” through cell culture-based studies. AMPK is widely recognized as crucial for the sensitive regulation of cellular energy metabolism, the regulation of ABCA1 expression, and the acceleration of cholesterol efflux from macrophages [31]. In this study, we not only validated the notion that AMPK activation is fundamental to MP's promotion of ABCA1 expression but also elucidated the cascade between AMPK activation (initial event) and the subsequent reduction in ABCA1 degradation (endpoint event) (Figure 7). Given that ABCA1 is the key mediator facilitating macrophage cholesterol efflux, constituting the first and rate-limiting step in RCT [36,37], the findings presented here may suggest a potential candidate molecule for cholesterol-lowering drugs targeting ABCA1. However, several issues still require attention in future research. For instance, (1) current research on the biological effects of MP is limited to the cellular level, and in vivo studies are necessary to further characterize its performance; (2) the mechanism by which MP inhibited the interaction of calpain with ABCA1 differs from the previous conclusion that promoted the dephosphorylation of PEST sequences [20], and there may be a new mechanism that remains to be explored.

In summary, we have developed an AMPK agonist, MP, that not only inhibits extracellular Ca^2+^ influx by targeting the Cav1.2 channel but also lowers intracellular Ca^2+^ concentration. This, in turn, inhibits calpain-mediated ABCA1 degradation. Additionally, it weakens the interactions between calpain and ABCA1, as illustrated in Figure 7. Collectively, these effects synergistically contribute to the reduction in ABCA1 degradation.

## 4. Materials and Methods

### 4.1. Reagents and Antibodies

MP was provided by the Institute of Material Medica, Chinese Academy of Medical Sciences (higher than 99.5% purity by HPLC). Reagents for cell culture, including Dulbecco’s modified Eagle’s medium (DMEM), RPMI 1640 medium, phosphate buffer saline (PBS) buffer (1×), penicillin-streptomycin mix (100×), and phorbol 12-myristate 13-acetate (PMA) were purchased from Beijing Solarbio Science & Technology Co., Ltd.(Beijing, China). Dimethylsulfoxide (DMSO) and Bovine Serum Albumin (BSA) were purchased from Sigma. Fetal bovine serum (FBS) and trypsin were purchased from Corning Incorporated (Shanghai, China). TRIzol reagent was obtained from ThermoFisher Scientific (Waltham, MA, USA). Reagents for preparation of recombinant full-length AMPK heterotrimeric complexes, including agar, isopropyl-beta-D-thiogalactopyranoside (IPTG), ethylene diamine tetraacetic acid (EDTA), protease inhibitor cocktail, and coomassie brilliant blue powder (CBB) were provided from Beyotime Biotechnology (Shanghai, China). Tryptone and yeast extract were purchased from ThermoFisher Scientific (Waltham, MA, USA). Sodium chloride, imidazole, and phenylmethanesulfonyl fluoride (PMSF) were purchased from Shanghai Macklin Biochemical Technology Co., Ltd. (Shanghai, China). Glycerol, chloropropanol, ethanol, and acetic acid were purchased from Beijing InnoChem Science & Technology Co., Ltd. (Beijing, China). Ampicillin and chloramphenicol were purchased from Shanghai Aladdin Biochemical Technology Co., Ltd. (Shanghai, China). Tris(2-carboxyethyl) phosphine hydrochloride (TCEP) was obtained from Beijing Solarbio Science & Technology Co., Ltd. (Beijing, China). Deoxyribonuclease I (DNase I) was provided from Gene-Protein Link Biotech (Beijing, China). Tris•Hcl (pH 8.0) was purchased from M&C Gene Technology (Beijing), LTD. (Beijing, China) Reagents for Electrophysiological recordings, including zeocin, G418, and hygromycin B, TEA-Cl, CsCl, EGTA, Na_2_-GTP, Mg-ATP, CsOH were provided from ICE BioScience Co., Ltd. (Beijing, China). Inhibitors, including Compound C, SKF96365, BAPTA-AM, 2-Aminoethyl Diphenylborinate (2-APB), and MG-132 were purchased from Selleck Chemicals (Shanghai, China). Ionomycin was obtained from Beyotime Biotechnology (Shanghai, China). Reagents for phosphorylation sites analysis, including DL-dithiothreitol (DTT), iodoacetamide (IAM), formic acid (FA), acetonitrile (ACN), methanol, were purchased from Sigma (St. Louis, MO, USA). Protease was purchased from Promega (Madison, WI, USA). Ultrapure water was prepared from a Millipore purification system (Billerica, MA, USA).

The list of antibodies is as follows, anti-ABCA1 (ab18180, Abcam, Cambridge, UK), anti-Calpain (MA3-940, ThermoFisher Scientific, Waltham, MA, USA), anti-AMPK (D63G4, CST, Danvers, MA, USA), anti-p-AMPK (D4D6D, CST), anti-LXRα (BM5134, Boster, CA, USA), anti-β-actin (K200058M, Solarbio, Beijing, China), anti-GAPDH (D16H11, CST), and anti-ABCA1 for Co-IP (ab307534, Abcam). HRP-conjugated secondary antibodies (A0216 and A0208, Beyotime, Shanghai, China).

### 4.2. Cell Culture

J774A.1 and THP-1 cells were provided from the BeNa Culture Collection. J774A.1 cells were cultured in DMEM plus 10%FBS, penicillin (100 U/mL), and streptomycin (100 U/mL). THP-1 cells were cultured in suspension in RPMI 1640 medium supplemented with 10% FBS. PMA (100 ng/mL) was used to induce THP-1 cells to differentiate into macrophages. J774A.1 and THP-1 cells were maintained in a 37 °C incubator with 5% CO_2_. Cells used for the following experiments were 5–9 passages, reaching 80–90% confluence. All experiments were repeated three times.

### 4.3. Cell Viability Assay 

The J774A.1 cells were equally seeded in a 96-well plate (1 × 10^5^ cell/mL) and cultured for 24 h. After that, cells were treated with different concentrations (1, 2.5, 5, 10, 25, 50, 100, 200, 400 μM) of MP for 24 h to explore the suitable intervention concentration. Then, their viability was evaluated using CCK-8 (Gene-Protein Link Biotech, Beijing, China). Briefly, 100 μl serum-free medium plus 10% CCK-8 solution was added to each well and incubated for 2 h at 37 °C. Finally, the absorbance was detected at 450 nm using a microplate reader, and cell viability was calculated as follows:Cell viability (%) = (OD_sample_ − OD_blank_)/(OD_control_ − OD_blank_) × 100%(1)

### 4.4. Preparation of Recombinant Full-Length AMPK Heterotrimeric Complexes

#### 4.4.1. *Escherichia coli* Culture and Protein Expression

The expression plasmid of full-length Rattus norvegicus AMPKα1β1γ1 was constructed in our previous work. A 6×His tag was introduced at the N-terminal end of the α-subunit to facilitate purification. The expression plasmid was transformed into Transetta (DE3) chemically competent cells (TransGen Biotech Co., LTD. Beijing, China), and transformants were selected on LB agar plates containing ampicillin (100 μg/mL) and chloramphenicol (33.3 μg/mL). Single colonies were used to inoculate 5 mL of LB medium containing 100 μg/ml ampicillin and 33.3 μg/mL chloramphenicol. A conical flask containing 1L LB medium supplemented with 100 μg/mL ampicillin and 33.3 μg/mL chloramphenicol was inoculated with 2 mL of the above overnight culture for large-scale expression and purification. Then, the Escherichia coli was grown in a shaker incubator with 220 rpm at 37 °C until the A_600_ reached 0.6. Protein expression was induced by 500 μM IPTG. At the same time, the incubator temperature was reduced to 16 °C, and the Escherichia coli was grown for an additional 18 h.

#### 4.4.2. Protein Extraction

After expression, Escherichia coli was harvested and resuspended in 10 ml lysis buffer [20 mM Tris-HCl (pH 8.0), 150 mM NaCl, protease inhibitor cocktail, 1 mM PMSF, 5% glycerol]. The Escherichia coli suspension was sonicated on ice for 2 min at 50% power. Then, 100 μL DNase I was put in the suspension and placed on ice for 30 min. Insoluble material was removed by centrifugation at 18,000× *g* in a Backman centrifuge for 30 min at 4 °C.

#### 4.4.3. Protein Purification

The supernatant was filtered using a 0.45 μm filter and then loaded on 5 mL His-tag protein purification beads (Solarbio, Beijing, China). Then, wash buffer [20 mM Tris-HCl (pH 8.0), 150 mM NaCl, 100 mM imidazole, protease inhibitor cocktail, 1mM PMSF, 5% glycerol] was used to wash five times to remove the non-specifically bound substances. After that, a gradient elution procedure was applied to obtain the expected protein. The detailed imidazole elution concentrations were 150 mM, 200 mM, 300 mM, 400 mM, 500 mM, 500 mM, and 500 mM, elution buffer volume was 10 mL. Subsequently, fractions containing the expected protein were pooled based on 8% SDS-PAGE and CBB analysis and concentrated to 1 ml using an ultrafiltration tube.

Apart from affinity chromatography, we used HiLoad 16/600 Superdex 200 pg (Cytiva life sciences, Logan, UT, USA) to purify the target protein further. Elution buffer [20 mM Tris-HCl (pH 8.0), 150 mM NaCl, 1 mM EDTA, protease inhibitor cocktail, 0.5 mM PMSF, 1.5 mM TECP, 5% glycerol] was used to elute the expected protein, and elution volume was 180 mL. Fractions were collected manually according to the A_280_ signal and then pooled based on 8% SDS-PAGE and CBB analysis. Lastly, the protein was concentrated using an ultrafiltration tube and then measured the final concentration, stored in a −80 °C refrigerator.

### 4.5. MST Analysis

We used a Monolith^TM^ RED-NHS second-generation protein-labeling kit (Nano Temper, South San Francisco, CA, USA) to label the protein samples. First, 100 μL purified AMPK protein (10 μM) was buffer exchanged using the labeling buffer NHS and A-column. The protein was then labeled with RED-NHS dye (final concentration 300 μM), incubated at room temperature for 30 min in the dark, and the labeled protein was collected. The tagged protein was then purified using B-column and stored in a −80 °C refrigerator for later use. The specific operation was carried out according to the kit instructions.

For MST detection, MP was diluted into 16 concentration gradients, as follows: 400,000 nM, 200,000 nM, 100,000 nM, 50,000 nM, 25,000 nM, 12,500 nM, 6250 nM, 3125 nM, 1562.5 nM, 781.25 nM, 390.63 nM, 195.31 nM, 97.66 nM, 48.83 nM, 24.41 nM, 12.21 nM. After that, the above MP work solutions were co-incubated with tagged AMPK protein (50 nM) at room temperature for 5 min in the dark, and then samples were loaded in the capillary for subsequent detection. MST data were collected by Mo Pico Microscale thermophoresis (Nano Temper, South San Francisco, CA, USA) and analyzed using MO Affinity Analysis software (3_v01).

### 4.6. ITC Analysis

Microcal PEAQ-ITC system was employed to reveal whether there was a direct interaction between MP and AMPK and the accompanying thermodynamic parameters. All experiments were conducted at 298.15 K, with 19 injections (the first injection volume was 0.4 μL, and the rest were 2 μL). The concentrations of AMPKα1β1γ1 and MP were 20 μM and 400 μM, respectively. The detailed experimental procedures were performed according to the Instrument program. The K_d_, stoichiometry (N), ΔH, and ΔS were calculated from raw data by MicroCal PEAQ-ITC analysis software (v1.41). In addition, the ΔG was calculated using the following equation:ΔG = ΔH − TΔS(2)

### 4.7. Nuclear Magnetic Resonance (NMR) Analysis

#### 4.7.1. STD-NMR Experiments

MP and adenosine monophosphate (AMP, TopScience, Shanghai, China) were dissolved in DMSO to obtain a working solution at a concentration of 250 μM. AMPK was prepared in 0.01 M PBS buffer (pH = 7.0) with 10% D_2_O to give a working solution at a concentration of 7.5 μM. The complex ligand-protein molar ratio was 33.3:1. Samples were subjected to ^1^H NMR, water suppression ^1^H NMR, and STD NMR experiments at 25 °C.

#### 4.7.2. Titration Experiments

Titration experiments were applied to explore whether there was a competitive relationship between AMP and MP binding to AMPK. The concentration of MP was fixed at 250 μM, and the concentration of AMP increased sequentially (31.25, 62.5, 125, 250, 500, 1000, 2000, 4000 μM).

#### 4.7.3. Data Collection and Processing

STD-NMR experiments were performed on Bruker Avance 500 MHz NMR spectrometer (Swiss) using the Topspin 3.6.2 suite. The experimental parameters were as follows: pulse sequence was SCREEN_STD, spectral width was 16 ppm, central frequency (o1p, ppm) was 4.7 ppm, 0.73 ppm (off resonance), −40 ppm (on resonance), both relaxation delay and saturation time were set to 2 s, spinlock time (D29, msec) was set to 25 ms. All spectrograms were processed in Topspin 3.6.2 suite, and the STD NMR of protein-ligand was subtracted from the STD NMR of ligand and protein to obtain saturation transfer double difference (STDD NMR).

### 4.8. CD Spectroscopy Study

CD spectra were obtained from a JASCO-715 spectropolarimeter (Jasco International Co., Ltd., Tokiwa-machi, Chuo-ku, Japan) using a 1 mm path-length quartz cuvette. Experimental settings were set as follows: Measure range: 190–300 nm; Data pitch: 0.5 nm; Sensitivity: Standard; D.I.T.: 1 sec; Bandwidth: 1.00 nm; Start Mode: Immediately; Scanning Speed: 50 nm/min; CD Detector: PMT; PMT Voltage: Auto. The baseline was obtained using 0.4 μM AMPK solution in 40 mM phosphate buffer (containing 10 mM NaF, pH 7.4). CD titration experiments were performed at a fixed concentration of AMPK (0.4 μM) incubated with MP at increasing concentrations in phosphate buffer (containing 10 mM NaF, pH 7.4). AMPK/MP molar ratios were 1:1, 1:5, 1:10, and 1:20. Each spectrum was obtained by averaging three scans and subtracting the contribution from corresponding blanks. The results were given in mdeg. The percentages of secondary structures were performed utilizing the CD analysis software (v2.12.00 [Build 1]), and secondary structure changes of AMPK at different concentrations of MP were distinguished.

### 4.9. Total and Membrane Protein Extraction

Total proteins from J774A.1 and THP-1 cells were extracted using RIPA lysis buffer (Gene-Protein Link Biotech, Beijing, China) supplemented with protease and phosphatase inhibitors cocktail (Roche ACCU-CHEK Active, China), with detailed procedures as described in previous work [38]. Cell membrane proteins were extracted using a membrane and cytosol protein extraction kit (Beyotime, Shanghai, China). Briefly, after washing with PBS, the cells were harvested by centrifugation at 600× *g* for 5 min at 4 °C. Then, 500 μL of reagent A (with 1 mM PMSF) was added, and the cells were placed on ice for 15 min. Cells were sonicated for 20 s on ice, and the supernatant was collected by centrifugation at 700× *g* for 10 min at 4 °C. Subsequently, the above supernatant was centrifuged at 14,000× *g* for 30 min at 4 °C to pellet cell membrane debris. Add 120 μL of reagent B to the tube to resuspend the pellet, and then place it on ice for 10 min. Finally, the cell membrane proteins were collected by centrifugation at 14,000× *g* for 5 minutes at 4 °C. All extracted proteins were assayed for concentration using the BCA protein assay kit (ThermoFisher Scientific, Waltham, MA, USA) and then stored at −80 °C for later analysis.

### 4.10. Quantitative RT-PCR Analysis

Total RNA in J774A.1 and THP-1 cells were extracted using TRIzol reagent. The detailed experimental procedures according to our previous work [23]. Primers used in current experiments were designed by PrimerBank and checked by the oligonucleotide property calculator, as follows: mouse ABCA1: GCCTGGATCTACTCTGTCGC (Forward) and GCCATTGTCCAGACCCATGA (reverse), human ABCA1: CTCGGTGCAGCCGAATCTAT (Forward) and CACTCACTCTCGCTCGCAAT (reverse), LXRα: AGAGTCTTGGGTCGCCAGTA (forward) and CTGGAGCCCTGGACATTACC (Reverse), β-actin: GAGCACAGAGCCTCGCCTTT (forward) and TCATCATCCATGGTGAGCTGG (reverse). The transcript abundance was normalized to the β-actin mRNA level.

### 4.11. ABCA1 Degradation Analysis

To investigate whether MP can enhance the stability of ABCA1, the degradation rate of ABCA1 was analyzed in the current experiment. Briefly, THP-1 cells were equably seeded in a 3.5 cm dish (total 5 × 10^5^ cells) and pretreated with either vehicle (RPMI medium with 1% BSA) or 100 μM MP for 24 h. After that, cells were washed with 1 × PBS and then maintained in 1% BSA medium with 20 μg/mL CHX (purchased from CST) for the indicated periods (15, 30, 60, 120, 240 min). The degradation of ABCA1 at various time points was examined by immunoblotting.

### 4.12. Calpain-Mediated Degradation Analysis

To confirm that ABCA1 can be degraded by calpain-mediated proteolytic pathway, THP-1 macrophages were treated with CHX plus increasing Ca^2+^ (0, 1, 10, 100, 1000, 2000 μM) with or without MP for 30 min after incubated with 10 μM BAPTA-AM for 3 h. The degradation of ABCA1 and the expression level of calpain were detected by immunoblotting.

### 4.13. Measuring Intracellular Ca^2+^ Level

Intracellular Ca^2+^ level was detected using the Fluo-4 AM kit (Solarbio, Beijing, China). THP-1 macrophages were first loaded with 5 μM of Fluo-4 AM working solution and incubated at 37 °C in the dark for 20 min. Later, quintuple Hank’s Balanced Salt Solution (HBSS, Solarbio, Beijing, China) containing 1% FBS was added and incubated for another 40 min to ensure intracellular transformation of Fluo-4 AM into Fluo-4. After gently washing thrice using HEPES buffer saline (Solarbio, Beijing, China), the fluorescence intensity was quantified using an Enspire® multimode reader (PerkinElmer, Singapore).

### 4.14. Electrophysiological Recordings

Cells used for electrophysiological recordings were provided by ICE BioSci. CHO cells and HEK cells were stably expressing the Cav1.2 channel and Cav3.2 channel, respectively. CHO cells were maintained in HAM’S/F-12 medium containing 10% FBS, 100 µg/mL Zeocin, 800 µg/mL G418, and 200 µg/mL Hygromycin B. Before patch clamp experiments, cells were separated using 0.25% trypsin-EDTA and treated by tetracycline for 24–72 h. HEK cells were maintained in DMEM containing 10% FBS and 200 µg/mL Hygromycin B. Before patch clamp experiments, cells were separated using 0.25% trypsin-EDTA and cultured in a 24-well plate for 18 h. The extracellular solution for whole-cell current measurement contained 140 mM TEA-Cl, 2 mM MgCl_2_•6H_2_O, 10 mM CaCl_2_•2H_2_O, 10 mM HEPES, and 5 mM D-Glucose. The pH was adjusted to 7.4 using TEA-OH. The internal pipette solution contained 120 mM CsCl, 1 mM MgCl_2_•6H_2_O, 10 mM HEPES, 10 mM EGTA, 0.3 mM Na_2_-GTP, 4 mM Mg-ATP, the pH was adjusted to 7.2 using CsOH.

The whole-cell patch clamp recordings were conducted using an EPC-10 patch-clamp amplifier (HEKA, Stuttgart, Germany), MP-285 Micromanipulator (Sutter Instruments, Novato, CA, USA), and PatchMaster software (HEKA, Stuttgart, Germany, v2×92). The cell membrane voltage was clamped at −80 mV after forming a whole-cell seal. The clamp voltage was maintained from −80 mV depolarized to +10 mV for 0.3 s. The data acquisition was repeated at 20 s intervals to observe the effect of the MP on the peak Cav1.2 current. Experimental data were collected by the amplifier and stored in the PatchMaster software (HEKA, Stuttgart, Germany, v2×92). The inhibition ratio of MP on the Cav1.2 channel was calculated as follows:Inhibition ratio = 1 − (Peak current*_MP_*/Peak current*_vehicle_*)(3)

### 4.15. Co-Immunoprecipitation (Co-IP)

A commercial kit (absin) was applied for Co-IP experiments, and all procedures were performed according to the manufacturer’s instructions. In brief, protein lysates were collected from THP-1 macrophages and washed with 500 μL lysis buffer (containing 5 μL protein A/G agarose beads). The lysates were incubated with anti-ABCA1 antibody or control lgG overnight at 4 °C, followed by incubation with the protein A/G agarose beads for 7 h at 4 °C. Later, the pellets were collected, rinsed thrice with 1 × wash buffer, and resuspended in 1 × SDS loading buffer. The immunoprecipitated proteins were then eluted from the beads by incubation at 95 °C for 5 min. After centrifuging at 14,000× *g* for 1 min at 4 °C, the supernatant was obtained and stored at −80 °C for further detection.

### 4.16. Identification of Phosphorylation Modification Sites

#### 4.16.1. Sample Preparation and Analysis

In brief, protein samples of ABCA1 were first extracted through Co-immunoprecipitation and then separated in 8% SDS-PAGE. The target gel containing ABCA1 was cut for subsequent experiments, including decolorization, dehydration, alkylation, digestion (trypsin and chymotrypsin enzymolysis), peptide fragment extraction, and Nano LC-MS/MS Analysis. The detailed procedures and experimental parameters were described in Appendix A.

#### 4.16.2. Data Analysis

The raw MS files were analyzed and searched against the target protein database based on the species of the samples using Byonic. The parameters were set as follows: the protein modifications were carbamidomethylation (C) (fixed), oxidation (M) (variable), Phospho (S, T, Y) (variable), the enzyme specificity was set to trypsin or chymotrypsin; the maximum missed cleavages were set to 3; the precursor ion mass tolerance was set to 20 ppm, and MS/MS tolerance was 0.02 Da. Only highly confident identified peptides were chosen for downstream protein identification analysis.

### 4.17. Immunoblotting

The extracted proteins were separated by 8% sodium dodecyl sulfate-polyacrylamide gel electrophoresis (SDS-PAGE) to analyze the protein expression level. Proteins were then transferred to polyvinylidene difluoride (PVDF) membranes and blocked by 5% non-fat milk for 1 h at room temperature. After that, the whole membrane was rinsed with TBST to remove residual milk. Subsequently, bands of target proteins with large differences in molecular weight were cut and incubated with the corresponding primary antibodies, e.g., ABCA1 (254 kDa, 1:250 dilution), Calpain (80 kDa, 1:500 dilution), LXRα (51 kDa, 1:1000 dilution), β-actin (42 kDa, 1:3000 dilution), or GAPDH (37 kDa, 1:2000 dilution). After that, all bands were incubated with the appropriate HRP-conjugated secondary antibody and visualized using ECL reagents (Solarbio). 

In addition, for target proteins with very small differences in molecular weight, e.g., AMPK (62 kDa, 1:1000 dilution), p-AMPK (62 kDa, 1:1000 dilution). After visualizing one of the target proteins, we incubated the bands for 30 min at room temperature on a shaker using a stripping solution to remove the bound antibody and re-performed the incubation with the blocking as well as the other antibody. 

Finally, we visualized several small membranes cut from the same membrane separately and quantified the gray values of the protein bands to statistically analyze the relative protein expression levels (target protein/β-actin or GAPDH).

### 4.18. Statistical Analysis

All the abovementioned experiments were repeated three times for reproducibility. Data were processed using GraphPad Prism 8.3.0, and results were displayed as mean ± SD or mean ± error. Student’s t-test evaluated differences between two groups, and differences between three or more groups were evaluated by one-way analysis of variance (ANOVA). Dunnett’s multiple comparisons test and non-parametric tests were utilized in the present study when data were normally distributed or not normally distributed, respectively. Statistical significance was considered when *p*-value < 0.05. (* *p* < 0.05, ** *p* < 0.01, *** *p* < 0.001).

## 5. Conclusions

In summary, this study is a follow-up to the study on IMM-H007, which was the first to demonstrate that MP is indeed the functional form of IMM-H007, capable of directly activating AMPK with a moderate binding affinity. Throughout this energy release process, the spatial conformation of AMPK undergoes alteration, involving nearly all ligand protons on MP. Furthermore, we present a pioneering elucidation of the enhancing potential and pharmacological mechanism of MP on ABCA1 expression following AMPK activation. More specifically, activated AMPK downregulates the Cav1.2 channel to obstruct the influx of extracellular Ca^2+^, thereby diminishing intracellular Ca^2+^ levels, suppressing calcium-activated calpain activity. In addition, activated AMPK also weakens the interaction between calpain and ABCA1. This cascade of events culminates in the deceleration of calpain-mediated degradation of ABCA1 (Figure 7). Taken together, MP emerges as a direct AMPK agonist and a promising candidate compound for enhancing ABCA1 stability and boosting cellular cholesterol efflux. In studies to follow, we shall utilize structural biological studies to gain insight into the structure-function relationships in the activation of AMPK by MP and to delve into the potential mechanisms by which MP attenuates the interaction between calpain and ABCA1.

## Figures and Tables

**Figure 1 ijms-24-17280-f001:**
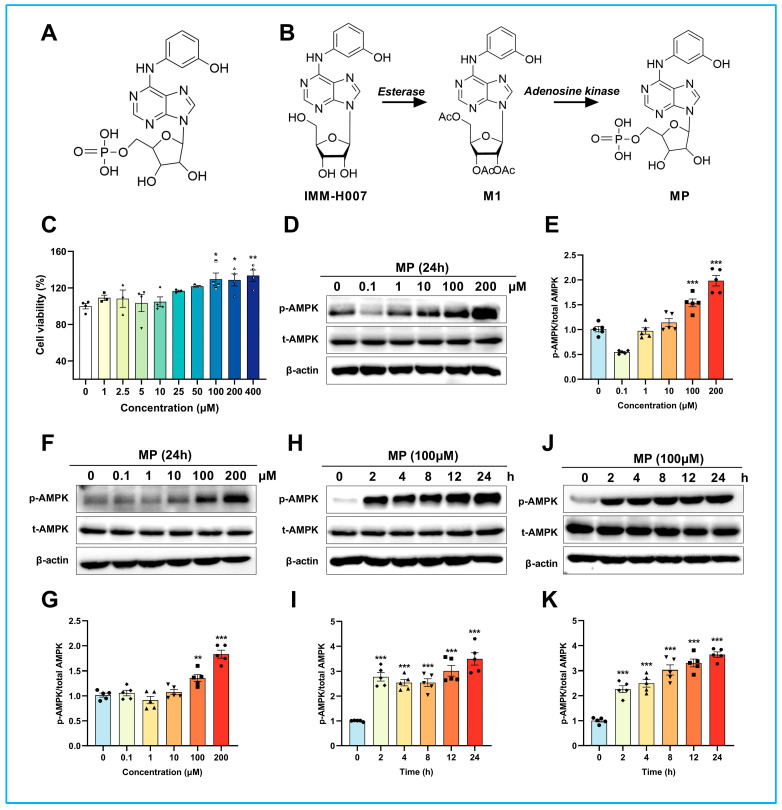
**MP facilitates AMPK Thr172 phosphorylation in a concentration- and time-dependent manner.** (**A**) The structure of MP. (**B**) The biotransformation of the prodrug IMM-H007 to MP. (**C**) Effects of different concentrations of MP on cell viability. (**D**) The relative expression of p-AMPK was increased concentration-dependent after treatment of J774A.1 cells with MP for 24 h. (**E**) The results of the quantitative analysis of (**D**). (**F**) The relative expression of p-AMPK was increased concentration-dependent after treatment of THP-1 macrophages with MP for 24 h. (**G**) The results of the quantitative analysis of (**F**). (**H**) The expression levels of p-AMPK were markedly increased in a time-dependent manner after treatment of J774A.1 with 100 μM MP. (**I**) The results of the quantitative analysis of (**H**). (**J**) The expression levels of p-AMPK were dramatically increased in a time-dependent manner after treatment of THP-1 macrophages with 100 μM MP. (**K**) The results of the quantitative analysis of (**J**). In (**C**), *n* = 6 for each group, data were analyzed using one-way ANOVA followed by Dunnett’s multiple comparisons and presented as mean ± SD. In (**D**–**K**), *n* = 5 for each group, the intensity of blots was measured using ChemiAnalysis software (CLINX, V17.12.08A). The actual data points for each group are shown as black symbols on the bar graph. Data were analyzed using one-way ANOVA followed by Dunnett’s multiple comparisons and presented as mean ± SD. * *p* < 0.05, ** *p* < 0.01 and *** *p* < 0.001 vs. the control group.

**Figure 2 ijms-24-17280-f002:**
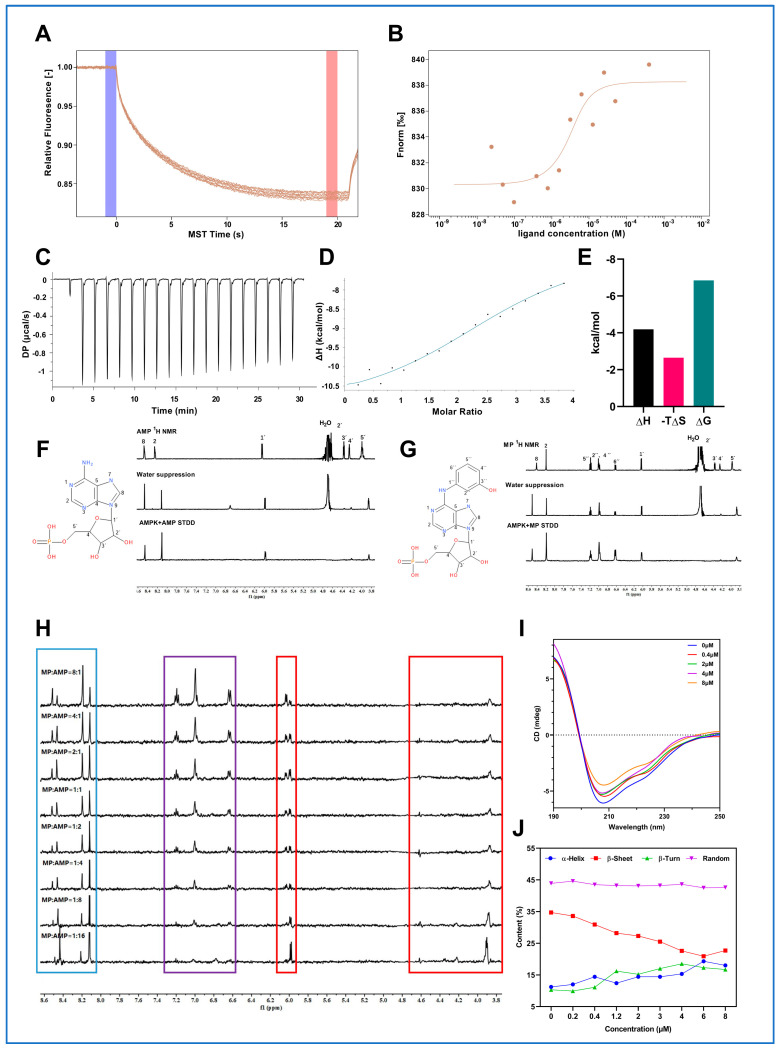
**Profile analysis of MP interaction with AMPK.** (**A**) The typical thermophoretic curve of MP binding to AMPK. (**B**) The binding curve of MP binding to AMPK (MST). (**C**) The titration curve of MP binding to AMPK. (**D**) The binding curve of MP binding to AMPK (ITC). (**E**) The concomitant thermodynamic alterations of MP binding to AMPK. (**F**) The binding epitopes analysis of AMP to AMPK. (**G**) The binding epitopes analysis of MP to AMPK. (**H**) The titration experiments for analyzing the competitive binding relationship between AMP and MP. (**I**) Far-UV CD spectra of MP binding to AMPK. (**J**) The secondary structure changes of AMPK after treatment with MP at multiple concentrations. All abovementioned experiments were repeated at least three times for reproducibility. Data were collected and processed by relevant software (specific software information was listed in the Section 4).

**Figure 3 ijms-24-17280-f003:**
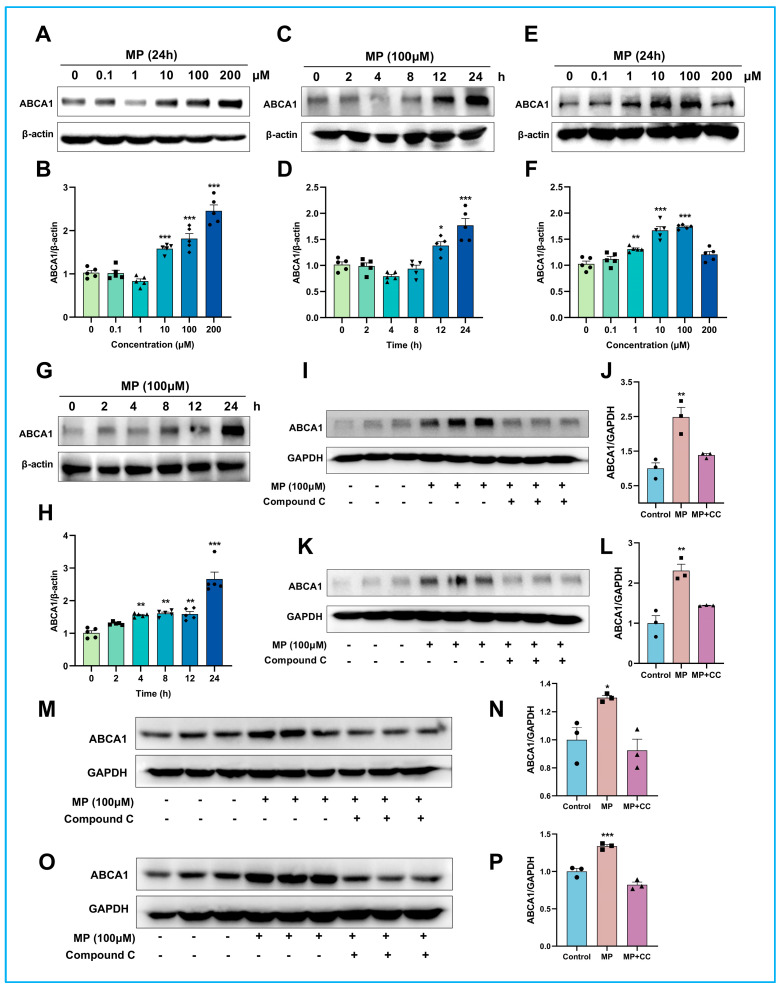
**MP promotes the expression of functional ABCA1 by activating AMPK.** (**A**) Effects of different concentrations of MP on the relative expression of ABCA1 in J774A.1 cells. (**B**) The results of the quantitative analysis of (**A**). J774A.1 cells were treated with different concentrations (0, 0.1, 1, 10, 100, 200 μM) of MP for 24 h. (**C**) Effects of treatment times with MP on the relative expression of ABCA1 in J774A.1 cells. (**D**) The results of the quantitative analysis of (**C**). J774A.1 cells were treated with 100 μM MP for different times (0, 2, 4, 8, 12, 24 h). (**E**) Effects of different concentrations of MP on the relative expression of ABCA1 in THP-1 macrophages. (**F**) The results of the quantitative analysis of (**E**). THP-1 cells were treated with different concentrations (0, 0.1, 1, 10, 100, 200 μM) of MP for 24 h. (**G**) Effects of treatment times with MP on the relative expression of ABCA1 in THP-1 macrophages. (**H**) The results of the quantitative analysis of (**G**). THP-1 cells were treated with 100 μM MP for different times (0, 2, 4, 8, 12, 24 h). (**I**) Compound C (5 μM) inhibition of AMPK abolished MP activation on total-ABCA1 in J774A.1 cells. (**J**) The results of the quantitative analysis of (**I**). (**K**) Compound C inhibition of AMPK abolished MP activation on total-ABCA1 in THP-1 macrophages. (**L**) The results of the quantitative analysis of (K). (**M**) Effect of MP on the expression level of functional ABCA1 in J774A.1 cells. (**N**) The results of the quantitative analysis of (**M**). (**O**) Effect of MP on the expression level of functional ABCA1 in THP-1 macrophages. (**P**) The results of the quantitative analysis of (**O**). In (**A**–**H**), n = 5 for each group, data were analyzed using one-way ANOVA followed by Dunnett's multiple comparisons and presented as mean ±  SD. In (**I**–**P**), *n* = 3 for each group, data were analyzed using one-way ANOVA followed by Dunnett’s multiple comparisons and presented as mean ±  SD. The actual data points for each group are shown as black symbols on the bar graph. The intensity of blots was measured using ChemiAnalysis software (CLINX, V17.12.08A). * *p* < 0.05, ** *p* < 0.01 and *** *p* < 0.001 vs the control group.

**Figure 4 ijms-24-17280-f004:**
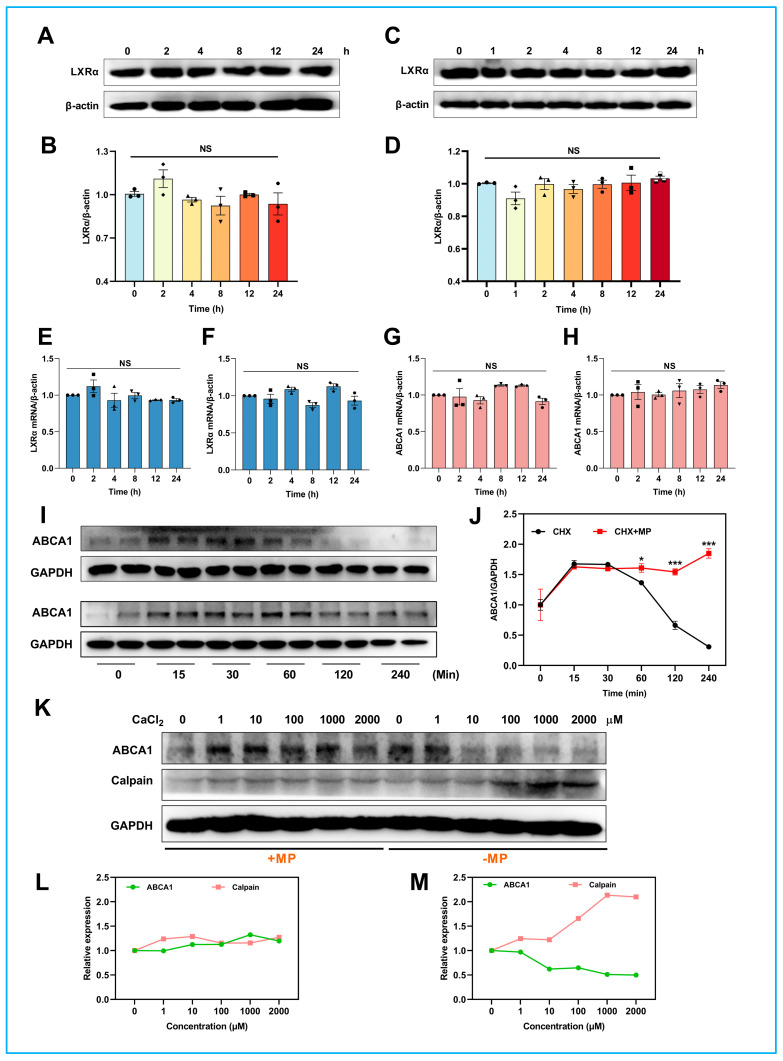
**MP promotes ABCA1 expression by inhibiting the calpain-mediated degradation pathway.** (**A**) MP did not affect the expression levels of LXRα in J774A.1 cells. (**B**) The results of the quantitative analysis of (**A**). J774A.1 cells were treated with 100 μM MP for different times (0, 2, 4, 8, 12, 24 h). (**C**) MP did not affect the expression levels of LXRα in THP-1 macrophages. (**D**) The results of the quantitative analysis of (**C**). THP-1 cells were treated with 100 μM MP for different times (0, 1, 2, 4, 8, 12, 24 h). (**E**,**F**) MP did not affect the LXRα mRNA levels. (**G**,**H**) MP did not affect the ABCA1 mRNA levels. (**I**,**J**) Effect of MP on ABCA1 degradation. (**K**) MP suppresses Ca^2+^-mediated ABCA1 protein degradation. (**L**,**M**) The results of the quantitative analysis of (**K**): (**L**) cells were exposed to various Ca^2+^ concentrations in the presence of 100 μM MP at 37 °C for 30 min; (**M**) cells were exposed to various Ca^2+^ concentrations in the absence of 100 μM MP at 37 °C for 30 min. In (**A**,**C**), *n* = 4–5 for each group, data were analyzed using one-way ANOVA followed by Dunnett's multiple comparisons and presented as mean ±  SD. In (**E**–**H**), *n* = 3 for each group, data were analyzed using one-way ANOVA followed by Dunnett’s multiple comparisons and presented as mean ± SD. In (**I**,**J**), *n* = 3 for each group, data were analyzed using Student’s t-test and presented as mean ±  error. The actual data points for each group are shown as black symbols on the bar graph. The intensity of blots was measured using ChemiAnalysis software (CLINX, V17.12.08A). NS: no significance, * *p* < 0.05 and *** *p* < 0.001 vs. the control group.

**Figure 5 ijms-24-17280-f005:**
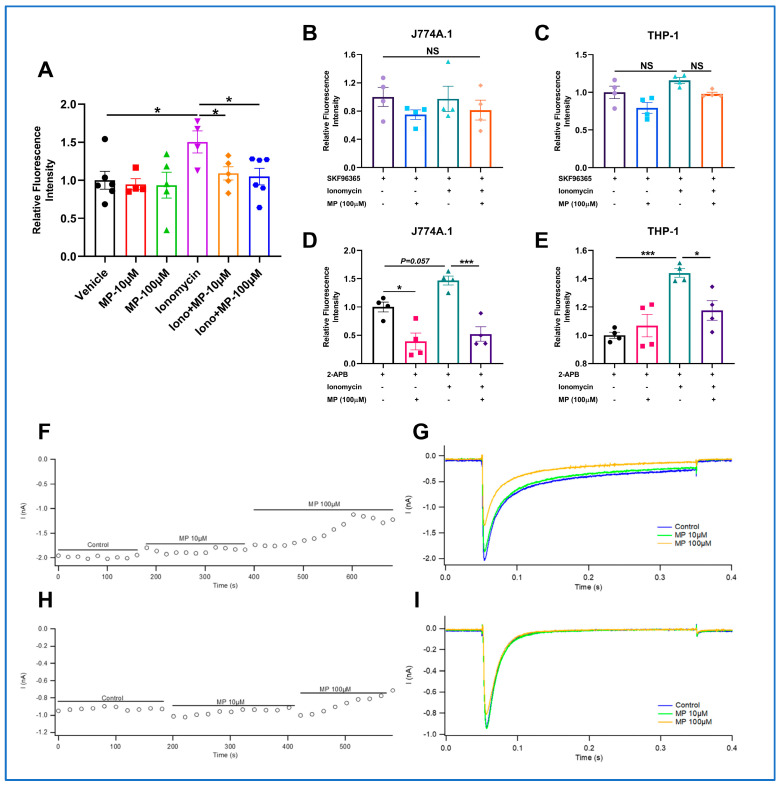
**MP suppresses the Cav1.2 channel to block extracellular Ca^2+^ influx and inhibit calpain activity.** (**A**) Effects of MP on the intracellular Ca^2+^ content. (**B**) Effects of MP on organelle Ca^2+^ efflux in J774A.1 cells. (**C**) Effects of MP on organelle Ca^2+^ efflux in THP-1 macrophages. (**D**) Effects of MP on extracellular Ca^2+^ influx in J774A.1 cells. (**E**) Effects of MP on extracellular Ca^2+^ influx in THP-1 macrophages. (**F**,**G**) Effects of MP on Cav1.2 channel. (**H**,**I**) Effects of MP on Cav3.2 channel. In (**A**–**E**), *n* = 3–4 for each group, data were analyzed using one-way ANOVA followed by Dunnett's multiple comparisons and presented as mean ±  SD. The actual data points for each group are shown as black symbols on the bar graph. The fluorescence intensity of (**A**–**E**) was quantified using an Enspire^®^ multimode reader. NS: no significance, * *p* < 0.05 and *** *p* < 0.001 vs. the control group.

**Figure 6 ijms-24-17280-f006:**
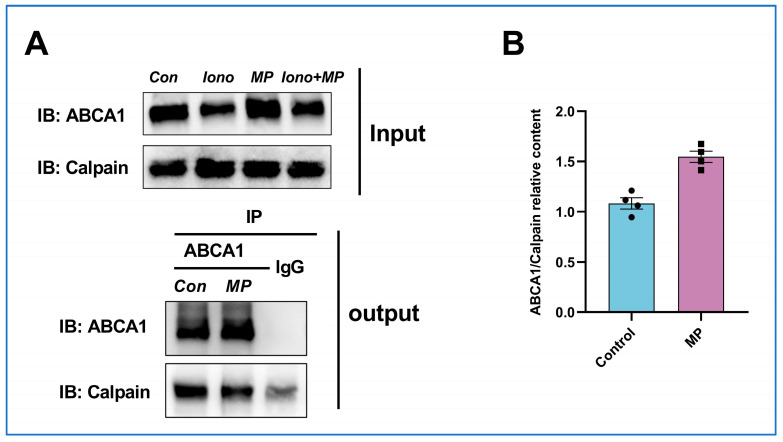
**MP attenuates the interaction between calpain and ABCA1.** (**A**) There was an interaction between calpain and ABCA1. (**B**) MP treatment attenuates the calpain-ABCA1 interaction. The actual data points for each group are shown as black symbols on the bar graph.

**Figure 7 ijms-24-17280-f007:**
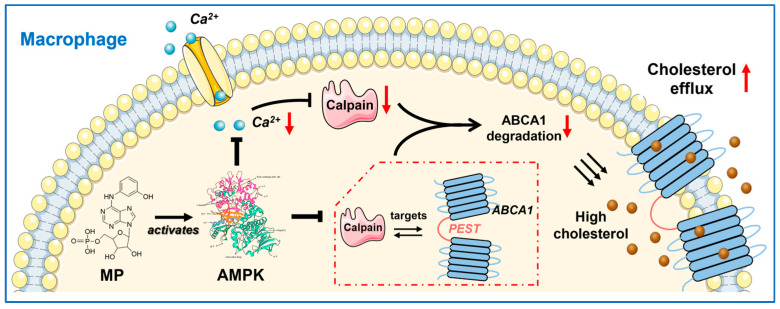
**Schematic diagram of the mechanism by which MP promotes cholesterol efflux from macrophages.** MP can directly bind to AMPK with a moderate affinity, alter its spatial conformation, and then stimulate AMPK Thr172 phosphorylation. After that, activated AMPK can repress the Cav1.2 channel to block extracellular Ca^2+^ influx, thereby reducing the intracellular Ca^2+^ level and inhibiting calcium-activated calpain activity. On the other hand, activated AMPK also weakens the interaction between calpain and ABCA1. Collectively, these actions delay calpain-mediated degradation of ABCA1 and promote cholesterol efflux from macrophages.

**Table 1 ijms-24-17280-t001:** Results of identification of phosphorylation sites on ABCA1.

Sample	Position	Peptide	Modification Type	Observed (M + H)	Score	Intensity
MP	1221	K.ELTKIY[+79.966]RR.K	Y [+80]	1158.6131	272.53	116,970,000
MP+Iono	1682	R.VS[+79.966]K.A	S [+80]	413.2074	44.73	2,667,500
	1114	K.LC[+57.021]C[+57.021]VGS[+79.966]SLFLK.N	S [+80]	1363.6768	43.31	9,350,100

## Data Availability

Data are contained within the article and Appendix A.

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
