# Peer review of "MP Allosterically Activates AMPK to Enhance ABCA1 Stability by Retarding the Calpain-Mediated Degradation Pathway"

_ijms, 2023, doi:10.3390/ijms242417280_

Round 1
Reviewer 1 Report
Comments and Suggestions for Authors
The manuscript by Li et al. analyzed the effect of AMPK agonist, MP, on the expression level/stability of ABCA1. As underlying molecular mechanisms, they focus on the regulation of intracellular Ca2+ level as well as Ca2+ activated protease, calpain. The first half of the studies examined biochemical and biophysical properties of MP. These analyses are important contents of the manuscript for these support the authors’ argument that previously reported atheroprotective effect of IMM-H007 is actually mediated by its in vivo derivative, MP. On the other hand, in the lest part of the study, although cell culture-based studies are employed, biological effect of MP is not clear. It could be pointed out that the lack of logical flow in the text as well as legend makes it difficult to understand which conditions are compared to each other to lead the expressions such as “MP promotes …”, “MP inhibits …” etc. Texts and legends related to Figure 3 onwards should be thoroughly revised.
Following points are examples which could be amended.
Fig 7: "weekens” does not make sense. Also, descriptions in the legend, relationships among MP, AMPK and calpain are misleading. The same applies to other parts of the manuscript where the word “dependent on” is used, e.g., section 2.5 etc.
Line 136: Names of softwares should be clarified.
Lines 213, 222, 287-289, 276-279, 296-311: they appear to be misleading or overestimations.
Figure legends: relationship between blot images and quantitation results shown as graphs is not clearly described. For example, in Fig 4, K, L, M could be better organized.
Figure 3I-3P: what is an interpretation on difference in the comparison results, total or membrane(=functional)?
Figure 5 and 6: It is questionable whether the comparisons performed on these results are rational (or citated properly in the text). The same question applies to section 2.8.
Results which seem to be performed only once need to be verified for their reproducibility (e.g., Fig. 4,5,6).
Table 1 is not properly designed (e.g., there are two entries in “Sample” raw, which is not consistent with the rest of the table).
Comments on the Quality of English LanguageIn some expressions, relationship among the molecules (MP, AMPK, ABCA1, calpain) is hard to understand. For example, in line 213, which is dependent on which (or which does what dependently on which) is not clear.
Author Response
Dear editor and reviewers,
On behalf of all the contributing authors, I would like to express our sincere appreciation for your letter and the constructive comments provided by the reviewers regarding our article titled "MP Allosterically Activates AMPK to Enhance ABCA1 Stability by Retarding the Calpain-Mediated Degradation Pathway (Manuscript ID: ijms-2673721)." These comments are invaluable and have greatly contributed to the improvement of our article.
In response to the comments from the editor and reviewers, we have undertaken extensive modifications to our manuscript. In this revised version, all changes have been highlighted within the document using red-colored text. Point-by-point responses to the reviewers are provided below this letter.
Reviewer
The manuscript by Li et al. analyzed the effect of AMPK agonist, MP, on the expression level/stability of ABCA1. As underlying molecular mechanisms, they focus on the regulation of intracellular Ca2+ level as well as Ca2+ activated protease, calpain. The first half of the studies examined biochemical and biophysical properties of MP. These analyses are important contents of the manuscript for these support the authors’ argument that previously reported atheroprotective effect of IMM-H007 is actually mediated by its in vivo derivative, MP. On the other hand, in the lest part of the study, although cell culture-based studies are employed, biological effect of MP is not clear. It could be pointed out that the lack of logical flow in the text as well as legend makes it difficult to understand which conditions are compared to each other to lead the expressions such as “MP promotes …”, “MP inhibits …” etc. Texts and legends related to Figure 3 onwards should be thoroughly revised.
Following points are examples which could be amended.
Fig 7: "weekens” does not make sense. Also, descriptions in the legend, relationships among MP, AMPK and calpain are misleading. The same applies to other parts of the manuscript where the word “dependent on” is used, e.g., section 2.5 etc.
A: Thank you for this thought-provoking comment. After careful examination, we agree with your point that this is indeed misleading. We have therefore revised Fig 7 and its legend to make the relationships among MP, AMPK and calpain clearer and more explicit. Also, all parts of the manuscript that contain “dependent on” have been checked and revised. Thanks again for your important comments!
Line 136: Names of softwares should be clarified.
A: Thanks for your kind reminder. All the softwares used in this study were checked and clarified in the revised manuscript.
Lines 213, 222, 287-289, 276-279, 296-311: they appear to be misleading or overestimations.
A: Thank you for pointing out these issues. Based on your suggestions, we have carefully revised these results descriptions to make them more detailed and proper. You can check it out in the revisions (marked in red), and please point out any incorrect revisions. Thanks again for your meticulous comments!
Figure legends: relationship between blot images and quantitation results shown as graphs is not clearly described. For example, in Fig 4, K, L, M could be better organized.
A: Thank you for your helpful advice. We have corrected all the figure legends that contain blot images and quantitation results to make them clearer and more detailed (marked in red). Thanks again for your careful comments!
Figure 3I-3P: what is an interpretation on difference in the comparison results, total or membrane(=functional)?
A: Thank you for kindly reminding us to explain this point. In Fig 3I and 3K, we investigated the effect of 100 μM MP on total cellular ABCA1 protein levels and whether this effect was associated with the activation of AMPK. As showed in Fig 3J and 3L, compared to the control group, total ABCA1 protein expression was significantly increased in the MP group, whereas this promotion was reversed by the addition of compound C (a specific inhibitor of AMPK), with MP exerting only a slight promoting effect. Since the extracted total ABCA1 protein contains both cell internalized ABCA1 and cell surface ABCA1, and the latter is referred to as functional ABCA1, it has been shown to be responsible for cholesterol efflux and HDL biogenesis [1]. Thus, we further studied the cell surface ABCA1 proteins individually to determine which ABCA1 is affected by MP and verify whether this effect is associated with the activation of AMPK (Fig 3M and 3O). Quantitative results were displayed in Fig 3N and 3P, compared to the control group, cell surface ABCA1 (functional) protein expression was significantly increased in the MP group, whereas this promotion was reversed by the addition of compound C. Taken together, MP promotes the expression of functional ABCA1 by activating AMPK. The relevant revisions in section 2.5 and legends (Fig 3) were marked in red. Please point out incorrect revisions (if any), thanks again for your comments!
Figure 5 and 6: It is questionable whether the comparisons performed on these results are rational (or citated properly in the text). The same question applies to section 2.8.
A: Thank you for your professional question. There were six groups (vehicle, MP-10 μM, MP-100 μM, ionomycin, ionomycin-MP-10 μM, ionomycin-MP-100 μM) in Fig 5A and 5B. The comparison between vehicle group and ionomycin group can verify whether the cells are modeled successfully and the confidence of the tool drug. The comparison between the ionomycin group and the ionomycin-MP groups can be used to test whether MP can reduce the excessive Ca2+ induced by ionomycin. For quantification, we equably seeded THP-1 macrophages in a 96-well black plate, and the fluorescence intensity of each group was quantified using an Enspire® multimode reader (PerkinElmer) after treatments. To visualize these results, we also used laser confocal microscopy to acquire fluorescence images of the cells and used them for presentation in the paper.
On the other hand, it is well known that either extracellular Ca2+ influx or organelle Ca2+ release causes an increase in intracellular Ca2+ content. Therefore, we used two inhibitors (SKF96365 for Ca2+ influx blocking and 2-APB for organelle Ca2+ release blocking) to separately block the above two pathways to understand the mechanism by which MP reduced intracellular Ca2+ content. It is foreseeable that the effect of MP on intracellular Ca2+ will disappear if the underlying mechanism of MP is blocked by the specific inhibitor (due to saturation of inhibition). Conversely, if the effect of MP on intracellular Ca2+ persists after pretreatment with the specific inhibitor, it indicates that MP does not act through the pathway blocked by the inhibitor (at least not primarily). In Fig 5C and 5D (results described in section 2.8), the intracellular Ca2+ level of SKF96365-MP group was slightly reduced compare to SKF96365 group, inferring that MP reduces intracellular Ca2+ level primarily by inhibiting extracellular Ca2+ influx and secondarily through organelle Ca2+ release. Meanwhile, the comparison between 2-APB-ionomycin group and 2-APB-ionomycin-MP group found that the lowering effect of MP on intracellular Ca2+ level persisted (Fig. 5E-G), again supporting the above view.
In Fig 6, the goal of this part is to investigate whether MP weakens the interaction between calpain and ABCA1. The immunoblotting results indicated that both ABCA1 protein and calpain were present in the whole cell lysate. In addition, calpain was still detected after co-immunoprecipitation with anti-ABCA1 antibody, suggesting that calpain interacts with ABCA1 protein. Moreover, it is predictable that if the strength of the interaction between calpain and ABCA1 protein is constant, then calpain should increase with increasing precipitated ABCA1. However, Further analysis found that the ABCA1 content increased in the MP group while the calpain content decreased, inferring that the MP could attenuate the interaction between calpain and ABCA1.
Finally, we have corrected the incorrect citation in the revised version. Thanks again for your helpful feedback. If there are any questions or incorrect revisions, please let us know and we will do our best to amend them.
Results which seem to be performed only once need to be verified for their reproducibility (e.g., Fig. 4,5,6).
A: Thank you for your important comment. We changed all the bar graphs to show the actual data points, and the repetitions within per group are all stated in the figure legends. All the experiments contained in Fig. 4, Fig. 5 and Fig. 6 were repeated three times for reproducibility.
Table 1 is not properly designed (e.g., there are two entries in “Sample” raw, which is not consistent with the rest of the table).
A: Thank you for give us the opportunity to clarify this confused point. In fact, in contrast to the normal group, our experiments identified phosphorylation sites in two groups, the MP group and the MP+Iono group. Among them, the MP group contained one phosphorylation site (1221) and the MP+Iono group contained two phosphorylation sites (1114 and 1682). Since table 1 is a three-line table, the internal frame lines are not visible, and it appears that the sample column has only two entries, while the remaining columns have three entries. To avoid this problem, we have adjusted the entry “MP+iono” to be top-center aligned in the revised version.
In some expressions, relationship among the molecules (MP, AMPK, ABCA1, calpain) is hard to understand. For example, in line 213, which is dependent on which (or which does what dependently on which) is not clear.
A: Thank you for your valuable comment. According to your comment, we have corrected the line 213 “MP promotes the expression of functional ABCA1 dependent on AMPK” to “MP promotes the expression of functional ABCA1 by activating AMPK”. Also, all parts of the manuscript that contain “dependent on” have been checked and revised. Besides, the relationship among the molecules (MP, AMPK, ABCA1, calpain) has been clearly and unambiguously shown in the revised Figure 7. Thanks again for your kind reminder!
We sincerely thank you again for your valuable feedback which would help to improve the quality of our manuscript!
Reference
[1] Wang, J., et al., Role of ABCA1 in Cardiovascular Disease. Journal of Personalized Medicine, 2022. 12(6).
Reviewer 2 Report
Comments and Suggestions for Authors
Li et al. studied how MP causes increased cholesterol efflux from two types of cultured cell models of human macrophage. Although they were able to find some new mechanistic insights, the study was directed by what are already proposed as possible mechanisms. As such, the progress made is largely incremental, and the nature of results presented is confirmatory. Some specific issues are described below.
1. The data on MP-AMPK binding are interesting, but I am not convinced that such binding occurs in the cell. The Kd for this binding is 5-10 µM. This indicates that in order for this complex to form in the cell, the concentration of both molecules must be in the order of micromole. This concentration may be possible for MP, but I doubt very much if AMPK concentration can reach this high. Please explain how the in vitro data are translatable into cells and also justify the proposed mechanism in this regard.
2. Fig. 5. The fluorescent images provided are meaningless because all I can see in the figure are black squares. Please provide figures that clearly show cells. It is not clear to me why the authors used confocal images for quantification. Wouldn’t it be better to use non-confocal images for this purpose? What does each dot in B through F represent; individual cells or averages of an experiment? If they are averages, it is not correct to take an average of averages. Please consult a statistician.
3. Discussion is a 100% repeat of introduction and results, which is not a discussion. Provide a meaningful discussion.
4. For all bar graphs, please superimpose actual data points as dots (like it is done in Fig. 5).
5. For quantification of immunoblotting data, how were the loading control bands obtained? Were they from the same membrane from which the data were obtained? Please describe how the loading control was done.
6. The original membranes of the immunoblotting provided are still partial (i.e. cropped). Please provide the whole membrane for each cropped data shown in figures.
7. All abbreviations must be defined. Units must be separated from numbers: for example, 10 mM, not 10mM.
Comments on the Quality of English Languagecareful editing is needed.
Author Response
Dear editor and reviewers,
On behalf of all the contributing authors, I would like to express our sincere appreciation for your letter and the constructive comments provided by the reviewers regarding our article titled "MP Allosterically Activates AMPK to Enhance ABCA1 Stability by Retarding the Calpain-Mediated Degradation Pathway (Manuscript ID: ijms-2673721)." These comments are invaluable and have greatly contributed to the improvement of our article.
In response to the comments from the editor and reviewers, we have undertaken extensive modifications to our manuscript. In this revised version, all changes have been highlighted within the document using red-colored text. Point-by-point responses to the reviewers are provided below this letter.
Reviewer
Li et al. studied how MP causes increased cholesterol efflux from two types of cultured cell models of human macrophage. Although they were able to find some new mechanistic insights, the study was directed by what are already proposed as possible mechanisms. As such, the progress made is largely incremental, and the nature of results presented is confirmatory. Some specific issues are described below.
- The data on MP-AMPK binding are interesting, but I am not convinced that such binding occurs in the cell. The Kd for this binding is 5-10 µM. This indicates that in order for this complex to form in the cell, the concentration of both molecules must be in the order of micromole. This concentration may be possible for MP, but I doubt very much if AMPK concentration can reach this high. Please explain how the in vitro data are translatable into cells and also justify the proposed mechanism in this regard.
A: Thank you for your helpful feedback. The equilibrium dissociation constant (Kd) of the ligand–receptor complex has been reported to be a very predictive parameter that links the in vivo concentrations with what might be expected pharmacodynamically at the receptor (when the concentration of ligand is equal to Kd then 50% of the receptors are occupied by the ligand) [1-2]. As such, we measured this basic parameter to evaluate the binding property of the drug-receptor. In this paper, the Kd of 1-10 µM means that MP can occupy 50% of the receptor (AMPK) by reaching this concentration range, rather than both ligand and receptor have to reach this concentration to produce binding. For instance, the experimental concentration of AMPK in the microscale thermophoresis test was 50 nM, well below the Kd value (1-10 µM), but MP binding to AMPK was still observed and the Kd value was calculated. In addition, the experimental concentration of AMPK in isothermal titrimetric calorimetric analysis was 20 μM, which correlates with its sensitivity for heat of binding detection. The instrument description recommends that the initial protein concentration be set at 20 μM and the ligand concentration be 10-20 times the protein concentration, with specific conditions based on pre-experiment results. On the other hand, our previous hamster pharmacokinetic results found that the maximum concentration of MP in blood exceeded 2000 ng/ml (>4.15μM) [3], which is in the range of Kd values, so that the binding of MP to AMPK is very likely to occur in vivo. Also, The MP distribution data in C57BL/6J mice (Data not yet published) showed that it was present in the liver and kidney at 1369.0 ± 280.9 (ng/g tissue) and 2692.0 ± 417.3 (ng/g tissue), respectively, and these concentrations were in the order of micromole, which again supported the above view.
- Fig. 5. The fluorescent images provided are meaningless because all I can see in the figure are black squares. Please provide figures that clearly show cells. It is not clear to me why the authors used confocal images for quantification. Wouldn’t it be better to use non-confocal images for this purpose? What does each dot in B through F represent; individual cells or averages of an experiment? If they are averages, it is not correct to take an average of averages. Please consult a statistician.
A: Thanks for your kind reminder, we will provide high-resolution fluorescence images in the revised version. For quantification, we equably seeded THP-1 macrophages in a 96-well black plate, and the fluorescence intensity of each group was quantified using an Enspire® multimode reader (PerkinElmer) after treatments. To visualize these results, we also used laser confocal microscopy to acquire fluorescence images of the cells and used them for presentation in the paper. Both above experiments were fluorescently labeled using the Fluo-4 AM kit (Solarbio). Since we repeated 4-6 wells per group in cell experiments to observe within-group differences and for statistical analysis, each dot in Fig 5B-F represents an actual data point in each group.
- Discussion is a 100% repeat of introduction and results, which is not a discussion. Provide a meaningful discussion.
A: Thank you for bringing this issue to our attention. We have rewritten the discussion part in the revised manuscript.
- For all bar graphs, please superimpose actual data points as dots (like it is done in Fig. 5).
A: Thank you for your valuable suggestion, we have changed all the bar graphs in the revised version.
- For quantification of immunoblotting data, how were the loading control bands obtained? Were they from the same membrane from which the data were obtained? Please describe how the loading control was done.
A: We would like to thank the reviewer for pointing out this issue. After a detailed examination of all loading control bands by the authors, we ascertained that the loading control bands are from the same membrane from which the data were obtained. The specific steps for obtaining the control bands and target protein bands were as follows.
Firstly, after blocking with 5% non-fat milk for 1 hour at room temperature, the whole membrane was rinsed with TBST to remove residual milk. Subsequently, bands of target proteins with large differences in molecular weight were cut and incubated with the corresponding primary antibodies, e.g., ABCA1 (254 kDa), Calpain (80 kDa), β-actin (42 kDa), or GAPDH (37 kDa). After that, all bands were incubated with the appropriate HRP-conjugated secondary antibody and visualized using ECL reagents.
In addition, for target proteins with very small differences in molecular weight, e.g., AMPK (62 kDa), p-AMPK (62 kDa). After visualizing one of the target proteins, we incubated the bands for 30 min at room temperature on a shaker using stripping solution to remove the bound antibody and re-performed the incubation with the blocking as well as the other antibody.
Finally, we visualized several small membranes cut from the same membrane separately and quantified the gray values of the protein bands to statistically analyze the relative protein expression levels (target protein / β-actin or GAPDH).
The above experimental procedures (Cut the whole membrane into several small membranes to conduct the subsequent experiments at the same time.) were implemented in this paper for the following reasons.
- It can avoid cross-interference between different antibodies and ensure the credibility of the experimental results.
- It can avoid the loss of target proteins on the PVDF membrane. (Since whole-membrane incubation can only detect one protein per experiment, multiple protein assays require multiple stripping, while repeated stripping will result in a loss of the content of the remaining proteins to be assayed, which may lead to failure of the experiment.)
- It can save experimental time. (The stripping steps requires multiple blocking and antibody incubation, which will take more time)
- It can save antibody dosage. (The small membrane incubation requires only 5 ml antibody, whereas whole membrane incubation requires 15 ml or more)
Even so, we are also aware that this experimental protocol may have the problem that the loading control and target bands do not come from the same membrane, therefore, all authors of this paper checked the immunoblotting graphs to ensure the accuracy of the experimental results. In addition, we will adopt the whole-membrane incubation protocol in our later work to avoid this problem. Thank you again for your professional advice!
- The original membranes of the immunoblotting provided are still partial (i.e. cropped). Please provide the whole membrane for each cropped data shown in figures.
A: Thank you for this valuable comment. As described in the fifth response, we cut the whole membrane into several smaller membranes after blocking so as to conduct the subsequent experiments at the same time. So, I am sorry that we cannot provide the whole membranes, but we can provide the cut intact smaller membranes upon your approval!
- All abbreviations must be defined. Units must be separated from numbers: for example, 10 mM, not 10mM.
A: Thank you very much for your kind reminder. We have scrutinized and corrected all the abbreviations and units.
We sincerely thank you again for your valuable feedback which would help to improve the quality of our manuscript!
Reference
[1] Terry P Kenakin., The optimal design of pharmacological experiments. In book: A Pharmacology Primer, 2022. (pp.269-306). DOI:10.1016/B978-0-323-99289-3.00014-2
[2] Ma, W.N., et al., Overview of the detection methods for equilibrium dissociation constant KD of drug-receptor interaction. Journal of Pharmaceutical Analysis, 2018. 3(8): p. 147-152.
[3] Jia, Y.F., et al., Simultaneous quantification of 2 ',3 ',5 '-tri-O-acetyl-N6-(3-hydroxylaniline) adenosine and its principal me-tabolites in hamster blood by LC-MS/MS and its application in pharmacokinetics study. Journal of Chromatography B-Analytical Technologies in the Biomedical and Life Sciences, 2016. 1022: p. 46-53.
Round 2
Reviewer 2 Report
Comments and Suggestions for Authors
1. The authors responded to my comments in an adequate manner. However, they failed to include these responses into the manuscript. It is likely that readers will have the same questions and concerns as this reviewer has initially, so it is best to incorporate your responses to my comments into the paper. For example, how the loading control was determined is an important point. Also add in discussion about the possible biological significance of the measured quantative MP-AMPK binding.
2. the fluorescent images are still too dark. I suggest deleting them altogehter as the quantification was done by using a plate reader. Modify the legend and methods accordingly.
3. The discussion ends abruptly, as though the authors has run out of breath. I suggest ending discussion with the statement and figure given as "Conclusions" at the very end of the text.
Comments on the Quality of English Language
Minor editing will help.
Author Response
Dear Reviewer,
We sincerely appreciate your valuable comments on our manuscript once again, which will significantly enhance the quality of our paper. Based on your nice suggestions, we have made detailed revisions to the current manuscript, and the specific modifications have been highlighted with a yellow background in the manuscript.
Please review the itemized responses below.
Comments and Suggestions for Authors
- The authors responded to my comments in an adequate manner. However, they failed to include these responses into the manuscript. It is likely that readers will have the same questions and concerns as this reviewer has initially, so it is best to incorporate your responses to my comments into the paper. For example, how the loading control was determined is an important point. Also add in discussion about the possible biological significance of the measured quantative MP-AMPK binding.
A: Thank you for your positive comments. In the revised version, we have incorporated detailed steps outlining how the loading control is obtained. Additionally, we have provided a thorough discussion on the significance of the MP-AMPK binding. Thanks again for bringing this to our attention!
- the fluorescent images are still too dark. I suggest deleting them altogehter as the quantification was done by using a plate reader. Modify the legend and methods accordingly.
A: Thank you for your suggestion. In the revised Figure 5, we have removed the fluorescent images and made corresponding modifications to the legends and methods.
- The discussion ends abruptly, as though the authors has run out of breath. I suggest ending discussion with the statement and figure given as "Conclusions" at the very end of the text.
A: Thank you for your professional and helpful suggestions. We have added a concluding statement at the end of the discussion to summarize our findings.
We sincerely thank you again for your valuable feedback!